# Maximizing Cumulative Trypsin Activity with Calcium at Elevated Temperature for Enhanced Bottom-Up Proteome Analysis

**DOI:** 10.3390/biology11101444

**Published:** 2022-10-01

**Authors:** Jessica L. Nickerson, Alan A. Doucette

**Affiliations:** Department of Chemistry, Dalhousie University, Halifax, NS B3H 4R2, Canada

**Keywords:** trypsin, calcium, cumulative activity, bottom-up proteomics, mass spectrometry, accelerated digestion, digestion efficiency, quantitative proteomics, sample preparation

## Abstract

**Simple Summary:**

Trypsin is frequently employed to cleave proteins ahead of mass spectrometry characterization. Traditionally, enzyme digestion involves overnight incubation of proteins at 37 °C, which is time consuming though still may yield poor digestion efficiency. While raising the temperature should theoretically accelerate the digestion, it also destabilizes the enzyme and promotes trypsin de-activation. We therefore questioned whether elevated temperature is beneficial for improving tryptic digestion. Here, we quantify protein digestion kinetics at elevated temperatures for calcium-stabilized trypsin and enforce the critical importance of calcium ions to preserve the enzyme. We quantitatively demonstrate that 1 h at 47 °C provides a superior digest when compared to conventional (overnight, 37 °C) processing of the proteome. The practical impact of our enhanced digestion protocol is shown through bottom-up mass spectrometry analysis of a complex proteome mixture.

**Abstract:**

Bottom-up proteomics relies on efficient trypsin digestion ahead of MS analysis. Prior studies have suggested digestion at elevated temperature to accelerate proteolysis, showing an increase in the number of MS-identified peptides. However, improved sequence coverage may be a consequence of partial digestion, as higher temperatures destabilize and degrade the enzyme, causing enhanced activity to be short-lived. Here, we use a spectroscopic (BAEE) assay to quantify calcium-stabilized trypsin activity over the complete time course of a digestion. At 47 °C, the addition of calcium contributes a 25-fold enhancement in trypsin stability. Higher temperatures show a net decrease in cumulative trypsin activity. Through bottom-up MS analysis of a yeast proteome extract, we demonstrate that a 1 h digestion at 47 °C with 10 mM Ca^2+^ provides a 29% increase in the total number of peptide identifications. Simultaneously, the quantitative proportion of peptides with 1 or more missed cleavage sites was diminished in the 47 °C digestion, supporting enhanced digestion efficiency with the 1 h protocol. Trypsin specificity also improves, as seen by a drop in the quantitative abundance of semi-tryptic peptides. Our enhanced digestion protocol improves throughput for bottom-up sample preparation and validates the approach as a robust, low-cost alternative to maximized protein digestion efficiency.

## 1. Introduction

Improved characterization of complex proteomic systems using enhanced MS instrumentation demands equally robust front-end workflows to fully capitalize on gains in sensitivity and throughput. For bottom-up proteomics, the trypsin digestion step [1] is integral to sample processing, though conventional enzyme digestion (37 °C, overnight, typically at a 50:1 protein to enzyme ratio) remains a bottleneck to maximize throughput. While adopting accelerated sample processing techniques is a shared priority in the field, protein digestion time cannot be optimized in isolation of other factors, including the efficiency of trypsin digestion, which directly influences quantitation accuracy, and analytical precision.

Multiple groups have reported enzyme digestion strategies to address sample processing throughput. Among these are the incorporation of chemically modified trypsin, which lowers autolysis [2,3,4], the use of high enzyme-to-substrate ratios including by way of immobilized enzyme reactors (IMER) [5,6,7], digestion at elevated temperatures or pressure [8,9], imparting energy by microwave radiation or ultrasonication [10,11,12,13,14], or the inclusion of sample additives such as organic solvents or surfactants [15,16,17]. These protocols have been employed to reduce the trypsin digestion time from hours to minutes, or even seconds. Recently, Zare’s group disclosed a sub-microsecond digestion strategy, enabled by enhanced trypsin kinetics within micron-sized droplets generated by electrosonic spray ionization [18]. As a metric of the accelerated digestion protocol, Zare’s study highlighted the improved protein sequence coverage afforded by the modified digestion strategy. However, we stress that the commonly cited result of generating more MS-detectable peptides does not fully reflect the extent of sample digestion, particularly when derived from single or simple protein mixtures. Perhaps counter-intuitively, improved sequence coverage can be obtained through incomplete digestion [19,20]. As shown by Cannon et al. in 2010, a middle-down approach (i.e., partial digestion) provides high sequence coverage of a ribosomal proteome [21]. Viewed from this perspective, elevated temperatures are also known to enhance trypsin de-activation [22,23,24,25,26,27], while even modest levels of organic solvents can induce protein (and enzyme) precipitation, contributing biased sample loss [28]. While surfactant-assisted digestion protocols exist (e.g., SDS, SDC), these detergents are known to significantly inhibit enzyme activity above threshold levels [29,30], and pose further challenges by their incompatibility with downstream processing [31,32]. Thus, despite the extensive options available, there is limited evidence to suggest these approaches offer a more effective digestion of the sample. Other critical parameters to assess include cleavage efficiency and enzyme specificity, each being relevant to modern proteome investigations.

The spectroscopic assessment of trypsin activity provides a preliminary estimate of digestion efficiency under a controlled set of conditions. A more complete understanding is provided by determining cumulative (integrated) trypsin activity over the full digestion period. To illustrate, while multiple researchers have noted enhanced enzyme activity at elevated temperatures [24,33,34], these gains will be tempered by accelerated enzyme de-activation [4,35], greater thermal aggregation of proteins [26], the potential for accelerated side modifications [36,37,38,39,40,41], and elevated chymotrypsin activity [42,43]. The ideal temperature for accelerated trypsin digestion presents a balance of enhanced initial activity with sustained trypsin stability over the duration of the incubation.

The influence of calcium ions to enhance tryptic activity and preserve thermal stability has been known since the early 1900′s [27,44,45,46,47], though surprisingly, the inclusion of calcium ions is not widely exploited across the proteomics community. The mechanism of calcium’s stabilizing effects was introduced by Gorini in 1951 [44] and expanded by Green and Neurath [48], whereby it was demonstrated that calcium decreases the rate of autolysis in both active and inactive forms of trypsin. A series of follow-up studies explored the effects of calcium on autolysis rates, conformational changes, and temperature dependence of trypsin-calcium interactions [27,45,49,50].

Given the extensive body of evidence to support a more active, stable, and selective enzyme, it was our objective to incorporate calcium-stabilized trypsin in an accelerated, high-temperature digestion workflow. Our interests are to adopt a routine and robust digestion protocol for accelerated proteolysis, delivering high specificity, and a high degree of cleavage which would in turn deliver high consistency. Through an optimized accelerated digestion protocol, the cleavage efficiency is assessed for a complex proteome by quantifying the extent of fully cleaved, fully tryptic and unmodified peptides, relative to those produced through a conventional digest. The merits of this robust, accelerated digestion workflow are discussed.

## 2. Materials and Methods

### 2.1. Trypsin Activity and Stability Assays

Initial trypsin activity was measured using benzoyl-arginine ethyl ester (BAEE) spectroscopic assays [51]. Briefly, 0.86 g/L BAEE substrate (Millipore Sigma, Oakville, ON, Canada) was combined with Tris (Thermo Fisher Scientific, Whitby, ON, Canada) adjusted to pH 8.0 and 1 μg TPCK-treated trypsin (cat. num. T1426, Millipore Sigma) in a temperature-controlled cuvette with or without calcium chloride (Millipore Sigma). The hydrolysis of BAEE was monitored by timed absorbance measurements at 253 nm across 2–3 min. The resulting slope (Δabsorbance/Δtime) was proportional to the enzyme’s activity. To determine enzyme stability over time, trypsin samples were buffered (pH 8) with or without calcium and aged at 37–67 °C for times ranging from 5 min to 16 h prior to combining with fresh BAEE substrate. At the indicated aging time, 1 μg of enzyme was sampled for BAEE assay and the resulting slope was normalized to that of the control (37 °C, no aging, no added calcium chloride). Control BAEE assays performed in the absence of trypsin confirmed the stability of the substrate at elevated temperatures.

### 2.2. Modeling Trypsin De-Activation Kinetics

The normalized residual activity was fitted using a second-order kinetic model. De-activation rate constants were extrapolated from the slopes of the linear correlation plot (inverse activity vs. time), and trypsin activity half-lives were calculated from Equation (1), where *k* is the de-activation rate constant and *A_0_* is the initial trypsin activity. The second-order models were also integrated to determine the cumulative enzyme activity across the measured time course.
(1)t12=1k A0

### 2.3. Bottom-Up Proteome Sample Preparation

*S. cerevisiae* was grown overnight at 30 °C in YPD broth (Millipore Sigma) to an OD_600_ of 1.0, as previously described [52]. Cell pellets were harvested by centrifugation, washed twice with distilled water, and lysed by grinding under liquid nitrogen. The lysate was subject to proteome extraction by boiling in 2% (*wt*/*vol*) SDS (Thermo Fisher Scientific) for 1 h. The resulting extract was isolated from residual cell debris by centrifugation. The total protein concentration was determined by a bicinchoninic acid (BCA) assay (Thermo Fisher Scientific) and adjusted to 2 mg/mL by dilution with water.

Aliquots of yeast proteome extract (100 μg) were subject to protein precipitation as previously described [53], by combining with 50 mM sodium chloride (Millipore Sigma) and 80% (*vol*/*vol*) acetone in the ProTrap XG filtration cartridge (Allumiqs, Halifax, NS, Canada). Following 2–5 min incubation with gentle mixing, protein pellets were isolated from the supernatant by centrifugation, washed with 400 μL additional acetone, and re-solubilized overnight in 8 M urea (Bio-Rad, Mississauga, ON, Canada). Solubilized proteins were diluted to 1.5 M urea with triethyl ammonium bicarbonate (TEAB) buffer (pH 8.0) (Millipore Sigma), and subject to reduction and alkylation with 5 mM DTT (Bio-Rad, Hercules, CA, USA) and 11 mM IAA (Millipore Sigma).

Trypsin digestion was performed at a 25:1 mass ratio (protein to trypsin) and with a variety of experimental conditions including the addition or absence of 10 mM calcium chloride, with a range of temperature (37 to 67 °C) and digestion times (15 min to 16 h). The conventional (control) digest consisted of overnight (16 h) incubation at 37 °C without calcium.

Following termination of the digests, all samples were subject to reductive dimethylation [54] with deuterated formaldehyde and sodium cyanoborohydride (Cambridge Isotope Laboratories, Tewksbury, MA, USA) for relative quantitation. Each experimental condition exhibited a “light” (+28.03 u) label while the conventional control digest was tagged “heavy” (+36.08 u). Experimental preparations were combined with the control at an equal mass ratio and subject to reversed phase LC-UV clean-up with total peptide quantitation from A_214_ and fraction collection in 45% acetonitrile. Desalted peptides were dried and stored at −20 °C until LC-MS/MS analysis.

### 2.4. Bottom-Up LC-MS/MS Data Acquisition

Bottom-up LC-MS/MS was conducted by injecting 1 μg total peptides onto a self-packed monolithic C18 column, coupled to a 10 μm New Objective PicTip noncoated Emitter Tip (Woburn, MA, USA). A Dionex Ultimate 3000 LC nanosystem (Bannockburn, IL, USA) delivered a 2 h linear gradient from 0.1% formic acid in water to 55% acetonitrile. The Q Exactive mass spectrometer (Thermo Fisher Scientific) operated in data dependent mode scanning the top 10 precursors for MS2, with a resolution of 35,000 full width at half maximum for both full MS and MS2.

### 2.5. LC-MS/MS Data Analysis

MaxQuant proteomics software version 1.6.17.0, developed by the Max Planck Institute of Biochemistry was downloaded from https://www.maxquant.org (accessed on 27 July 2022). Raw MS/MS spectra were searched in MaxQuant [54] with relative quantitation based on dimethyl labeling, first with full trypsin specificity and subsequently with semi-tryptic specificity, in both cases using an FDR of 0.01 and allowing up to 2 miscleavages. Search results were filtered at the peptide level, requiring a PEP of ≤0.01 and at the protein level, requiring ≥2 unique peptide IDs per protein identification. Peptide and protein identifications were compared across digestion conditions using Venny 2.1 [55]. Motif diagrams were generated using WebLogo [56].

### 2.6. Data Availability

Raw MS/MS spectra were searched in MaxQuant version 1.6.17.0 [54] with relative quantitation. The mass spectrometry proteomics data have been deposited to the ProteomeXchange Consortium (http://proteomecentral.proteomexchange.org (accessed on 27 July 2022)) via the PRIDE partner repository [57] with the dataset identifier PXD035682. 

## 3. Results

### 3.1. Cumulative Trypsin Activity Is Maximized at 47 °C with 10 mm Calcium Ions

Bottom-up proteomics workflows have exploited elevated temperatures for accelerated trypsin digestion (up to 90–100 °C) [58,59,60]. Our spectroscopic BAEE assays confirmed an optimal initial trypsin activity at 47 to 57 °C, respectively, contributing a 31 and 28% enhancement relative to the control, 37 °C (Figure 1). The addition of calcium not only promotes higher thermal stability for trypsin but also enhances its initial activity. Inclusion of 10 mM calcium chloride contributes a maximal 86% increase in initial activity at 37 °C. Moreover, 10 mM calcium shifts the optimal temperature for initial activity to 67 °C, whereby trypsin activity was observed to be 340% relative to the control condition. These results confirm the essential role of calcium ions to maximize tryptic activity at elevated temperature. However, such observations do not reflect the expected loss in enzyme activity caused by trypsin autolysis or thermal denaturation, meaning it cannot be concluded that 67 °C is optimal for tryptic digestion.

Enzyme stability studies were conducted by pre-incubating trypsin (pH 8) at various temperatures in varying concentrations of calcium prior to BAEE assays. Without calcium, the enhanced enzyme activity afforded by elevating the temperature is poorly sustained. After 30 min, the residual activity observed at higher temperatures had dropped to levels below that at 37 °C (Figure 2A). The loss of trypsin activity through time was consistent whether the enzyme was incubated alone or in the presence of a substrate protein (BSA, 25:1 ratio) (results not shown). Appendix A shows similar stabilization benefits from the inclusion of 5–100 mM calcium ions, with 10 mM minimizing the rate of deactivation. From Figure 2B, inclusion of 10 mM calcium chloride largely preserves the initial activity gains, though only to a maximum temperature of 57 °C. At 37 °C, enzyme de-activation was immeasurable across the first 4 h, with only a drop in activity after a 16 h incubation (Appendix A). The curves presented in Figure 2 represent best fit trendlines of the data using a second-order kinetic model (Appendix A). These fits allow for quantitative reporting of the rate of trypsin de-activation, as summarized in Table 1.

From Table 1, we note a 25-fold improvement in the stability of trypsin at 47 °C with calcium ions compared to the same temperature without calcium, as seen from the drop in the de-activation rate constant. Moreover, the enzyme shows a 5-fold increase in the half-life at 47 °C with calcium ions, at nearly 12 h, compared to only 2.4 h with conventional conditions (37 °C, no calcium). The stability improvement also adds to a >2-fold enhancement in initial activity at 47 °C. However, at 67 °C, while the initial activity was considerably higher, the presence of calcium no longer provided a stabilizing effect, resulting in a de-activation rate similar to the no-calcium condition.

To reflect trypsin activity over the duration of a digestion, we present the cumulative (integrated) enzyme activity (*A_T_*) as a function of temperature in Figure 3. These plots are obtained through the extrapolated rate constant and integrating the second-order kinetics curves to arrive at the equation below:(2)$$A_{T}=\frac{1}{k}ln\left(1+A_{0}kt\right)$$

In this equation, *k* is the de-activation rate constant, *A*_0_ is the initial trypsin activity and *t* is the cumulative digestion time. With inclusion of calcium, the cumulative activity is optimal at 47 °C at all points over a twelve-hour incubation period. It is also evident that higher temperature digests, specifically 67 °C, show a significantly lower cumulative activity, even relative to a 37 °C digestion. A temperature of 47 °C in the presence of 10 mM calcium was chosen as the preferable conditions for our enhanced rapid digestion approach. A direct comparison of the cumulative activity under this condition to that of a conventional digestion (37 °C, no calcium) is provided in Figure 3C. An overnight (16 h) digestion is seen to be equivalent to 2.5 h digestion under enhanced conditions. Likewise, a 1 h digest under enhanced conditions provides ~50% of the cumulative activity of a 12 h conventional digest. While these results suggest that accelerated digestion could achieve equivalent cleavage efficiency of a 37 °C digest but in a fraction of the time, it is realized that the BAEE assay does not capture the complexity of a proteomic system undergoing trypsin digestion. To determine the practical influence of these activity and stability enhancements, relative MS-based peptide quantitation was employed following bottom-up preparations ranging in digestion time, temperature, and calcium inclusion.

### 3.2. Qualitative Proteome Identifications Show High Similarity between Digestion Conditions

Figure 4 provides a summary of total peptide and protein identifications from each of the digestion conditions explored. Identifications are optimized in the 1 h digest at 47 °C with calcium, showing a 39% increase in peptide IDs compared to the conventional overnight digest. The Venn diagram in Appendix A shows that miscleaved peptide segments observed in the conventional preparation are often represented as more completely digested peptides in the rapid digest—this therefore increases the overall heterogeneity and non-redundant identifications. Without the addition of calcium, a 1 h digest at 47 °C resulted in 17% fewer peptide IDs and 26% fewer peptides than 1 h at 37 °C. This reflects the consequences of reduced enzyme stability at higher temperature without calcium (4-fold difference in enzyme half-life) and highlights the importance of conserving trypsin activity even across a 1 h digest. Moreover, under enhanced conditions, a 15 min digest produced 12% fewer peptides than the optimal 1 h digest. This suggests that valuable cleavage events are ongoing between the 15 min and 60 min points. The Venn diagrams in Figure 4B,C show 61.8% overlap in identified peptides and 94.2% protein overlap when comparing the enhanced 1 h digestion to the conventional overnight protocol. Only 6.8% of peptides (and no proteins) were unique to the conventional digest. Not only does the accelerated digestion protocol capture the proteome profile made visible through conventional digestion, but subsequent analysis of the resulting peptide profile suggests a more complete digestion of the sample using enhanced conditions.

We next assessed the proteome identifications based on miscleavage frequency to infer the degree of digestion completion (Figure 5). As expected, the frequency of fully cleaved peptides increases with longer digestion under conventional conditions, albeit with only marginal gains beyond 1 h. These “diminishing returns” are reflective of a decline in enzyme activity from 68% at 1 h to only 6% after 16 h (Figure 2).

Under enhanced conditions (47 °C with calcium), the frequency of fully cleaved peptides increases from 47.7% at 15 min to 56% at 60 min—equivalent to the overnight digest under conventional conditions (Figure 5). Both rapid digests at 47 °C also show 62% overlap with the peptides identified in the conventional sample (Appendix A); however, the higher frequency of fully cleaved peptides lends favor to a 1 h digest over the shorter 15 min protocol, which is attributed to there being four times as much cumulative activity. In the absence of calcium ions, the 1 h digest at elevated temperature had 15% higher miscleavage frequency (Figure 5), being comparable to 15 min at 37 °C, despite having more than double the cumulative activity (Figure 3).

### 3.3. Relative Quantitation Reveals Trypsin Cleavage Is Accelerated at Elevated Temperature with Added Calcium Ions

We next provide a quantitative examination of digestion products based on differential dimethyl labeling of various accelerated vs. conventional preparations. Peptides identified from each digestion condition were sorted based on their number of miscleavages and their cumulative signal intensities were plotted over time (Figure 6A,B). Under conventional conditions, longer digestions contribute a significant increase in the frequency of fully cleaved peptides, with a corresponding decrease in the intensity of miscleaved peptides, demonstrating the importance of overnight incubation to maximize cleavage under conventional conditions. At 47 °C with calcium ions present, lengthening the digest from 15 min to 1 h also yields a 10% increase in the intensity of fully cleaved peptides. However, extending the incubation period beyond 1 h does not yield a statistical gain in digestion efficiency.

The intensity ratio of peptides observed from four rapid digestion conditions were compared to those from conventional overnight digestion and grouped according to the degree of cleavage (fully cleaved, singly and doubly miscleaved) with results summarized in Figure 7. At 37 °C (A&B), shorter digests lead to fully cleaved peptides with a relative signal intensity below 1, while miscleaved peptides have ratios above 1. This translates to the overnight digestion having maximal signal intensity of fully cleaved peptides and minimizes signals for peptides with miscleavage sites. At 47 °C with calcium ions, we observe a more complete digest with 1 h incubation vs. 15 min (Figure 7C). Interestingly, we also note that a 1 h enhanced rapid digest increased the signal of all types of peptides compared to the conventional overnight digestion. The intensity ratio of fully cleaved peptides is well above 1, though so too are the miscleaved peptides, relative to conventional overnight digestion (Figure 7D). From this quantitative assessment, we show that the 1 h digest at 47 °C with calcium ions improves the overall throughput with enhanced digestion completion, while contributing greater intensity for peptides of all types compared to the conventional approach.

### 3.4. Trypsin Specificity Is Enhanced in the Rapid Digest at Elevated Temperature with Added Calcium Ions

The normally stringent specificity of trypsin has reportedly been compromised at elevated temperatures [42,43]. To evaluate the enzyme’s specificity in our enhanced rapid digest, the MS data were also subject to a search for semi-tryptic peptides. The enhanced and conventional digests demonstrated equivalent frequency of semi-tryptic cleavage, with 81% and 79% of identified peptides having full trypsin specificity, respectively. A Venn diagram comparing the semi-tryptic peptide identifications from the two digests (Figure 8A) shows 64% overlap, which is similar to that observed for fully tryptic peptides (Figure 4B). However, quantitative analysis of these semi-tryptic peptides shown in Figure 8B revealed a lower relative abundance of semi-tryptic peptides for the enhanced digestion compared to fully tryptic peptides, suggesting enhanced specificity is achieved in the rapid digest.

The cleavage efficiency of trypsin varies depending on the nature of the residues proximal to the cut site, with reduced digestion expected for dibasic sites and when K and R residues are adjacent to acidic residues [61,62]. We compared the relative cleavage efficiency of these “hard-to-cut” regions between the enhanced and conventional digests. From Figure 9A,B, it was found that 55% of all miscleaved peptides were contributed by acidic or dibasic cut sites. Based solely on the relative frequencies of miscleavage sites, the motif diagrams in Figure 9C reflect equivalent efficiency for digestion of acidic and basic amino acids within 2 residues or the C-terminal cleavage site. However, a quantitative assessment of these peptides (Figure 9D shows that acidic and dibasic C-terminal digestion sites were cleaved with significantly greater efficiency in the enhanced digest compared to the conventional (*p* < 0.001). The combination of enhanced cleavage of traditionally difficult sites with a reduction in semi-tryptic peptides lends a conclusion that the accelerated (1 h) incubation at 47 °C with added calcium ions provides an enhanced digestion relative to the conventional overnight (37 °C) approach.

## 4. Discussion

Proteolysis with trypsin is a critical, yet throughput-limiting component of bottom-up proteome workflows. Many works describe accelerated digests, which are typically validated by achieving high proteome and sequence coverage for simple protein systems. However, the growing interest of quantitative investigations motivates a high-throughput digestion approach that conserves the completion of a conventional overnight digest of a proteomic mixture. The present study first aimed to quantify both the initial activity and de-activation of trypsin over time as a function of temperature as influenced by the presence of calcium ions. Integrating the residual trypsin activity curves over extended periods provides a quantitative evaluation of the cumulative enzyme activity experienced over the duration of a digestion. Our results lead to a conclusion that the cumulative enzyme activity is maximized at 47 °C, justifying the use of this modestly higher temperature digestion but only in the presence of calcium ions. Performing digests at even higher temperatures is not justified as the enhanced initial activity seen in a BAEE assay is not sustained. For example, at 57 °C, the cumulative activity is at most ~50% above that obtained at 37 °C, but only for a short period. Owing to enzyme de-activation, cumulative activity drops below that seen at 37 °C after only 3 min. From the optimal digestion conditions suggested by these activity assays, quantitative bottom-up LC-MS was employed to compare accelerated protocols to a conventional (overnight, 37 °C digest). We evaluate not only the proteome coverage, but also the quantitative profiling of partially vs. fully cleaved peptides, as well as the digestion specificity.

Our evaluation of initial trypsin activity as a function of temperature showed strong agreement with the enhancement reported by Sipos et al. in 1973 [49]. They showed a 37% improvement in activity when temperature was increased from 37 °C to 47–57 °C, similar to the 30 ± 10% increase seen here (Figure 1B). Sipos also demonstrated that calcium ions shifted the optimal temperature to 67 °C owing to increased thermostability. Interestingly, though we observed the same temperature shift, we saw a greater enhancement factor (3.4 ± 0.2-fold compared to their 2.6-fold) (Figure 1B). This discrepancy may be attributed to lower enzyme purity in the earlier work. Likewise, the addition of calcium ions provided a 50% increase in initial activity at 37 °C, which was twice that observed by Sipos (~24%).

The stabilizing effects of calcium can sustain trypsin activity at higher temperatures for longer periods. However, at 67 °C, calcium no longer prevents trypsin deactivation. At elevated temperatures of 47 and 57 °C, we observed 19.3- and 5-fold increases in the enzyme’s half-life, respectively. Therefore, the addition of calcium ions provides a drastic improvement in trypsin stability so long as a critical temperature (57 °C) is not exceeded. This suggests that the stabilizing effects of calcium ions are limited by the rate of protein denaturation as opposed to the rate of autolysis at this temperature, which aligns with Trampari’s discussion of accelerated unfolding at similar temperatures [26]. The cumulative activity of trypsin we calculate by integrating the activity over time justifies the use of only a modest temperature increase, when in the presence of calcium ions. Considering a higher temperature digestion, the enhanced initial activity seen in a BAEE assay is not sustained, meaning cumulative activity suffers. For example, at 57 °C, the cumulative activity difference is maximized at very short times by ~50% vs. 37 °C. However, cumulative activity drops below that seen at 37 °C after only 3 min.

As reported by Nord and Bier (1956) [46], we found that trypsin de-activation initially followed second-order kinetics (Appendix A). Sipos reported a similar trend, noting a second-order rate for autolysis, with subsequent de-activation following a pseudo-first order rate [27]. We speculate that the gradual acceleration of activity loss is due to thermal aggregation, which is reported to occur most readily at temperatures exceeding 54 °C [26]. These trendlines allowed us to integrate the cumulative trypsin activity and show that a 1 h digest at 47 °C with calcium yields a cumulative enzyme activity equivalent to a 6 h digest under conventional conditions. However, applied to a real proteomic system, other benefits of high temperature digestion may be realized.

It is evident that short digestions yield many MS-detectable peptides. In fact, a 15 min incubation at 37 °C generated more peptides than the overnight control digestion, although with fewer total proteins identified. There was also a high degree of overlap (~95%) in the proteome coverage by each of the experimental digestion conditions compared to the control digest (Appendix A). Nonetheless, significant differences were observed. For example, shorter digestions translate into more miscleavages, which is evident from the data. Appendix A indeed shows a larger proportion of low-abundance proteins [63] identified using the optimized enhanced digestion protocol. Focusing on the types of peptides, as well as their relative abundances across the various digestion conditions allows a more complete assessment of each digestion condition. In theory, less complete digests yield a greater variety of peptides, which would be desirable for a single protein or simple proteome systems. However, for more complex systems, the variety of peptides generated would further mask lower abundance proteins from being detected. Appendix A shows that miscleaved peptides are not contributing significantly to the total proteome coverage, with 95% of protein identifications having at least one fully cleaved peptide in both the enhanced rapid and conventional digests. These results demonstrate the relevance of a robust (more complete) digestion for accurate and reproducible bottom-up quantitation.

We assume that a more complete digest will identify peptides with fewer miscleaved sites than a less complete digest and consequently more homogenous digestion products which lends favor to quantitative and targeted analyses. At 37 °C, fully cleaved peptides increase in relative intensity as one transitions from 15 min to 1 h to overnight digestion. This reflects a slower digestion rate compared to the enhanced conditions (47 °C with added calcium). Not only does the enhanced digestion produce fully cleaved peptides with higher relative intensity than the control digestion, but the intensity of peptides with miscleavage sites was also higher than that of the control. In other words, the enhanced digestion protocol provides a more complete digestion, but also yields higher signal intensities than the conventional digestion, meaning a greater number of peptides of all types can be detected. This may reflect a greater degree of protein or peptide loss during extended incubation. It may also be reflective of differences in the degree of peptide modifications over time. The enhanced digestion of the accelerated protocol motivates its application in bottom-up quantitative studies where complete and consistent production of proteotypic peptides is critical.

High temperatures have been reported to compromise the selective structure of the active site of trypsin. However, a qualitative assessment of semi-tryptic peptides showed that non-specific cleavage was less prevalent in the enhanced digestion protocol. Although the proportion of semi-tryptic peptides was equivalent to the control, quantitative analysis revealed a higher relative abundance of fully tryptic peptides compared to those with semi-specific cleavages. This suggests an enhancement in the enzyme’s specificity under the temperature- and calcium-enhanced conditions.

The cleavage efficiency of trypsin is highly dependent on the nature of the amino acid residue adjacent to the lysine or arginine. A 2005 report by Šlechtová et al. described differential cleavage rates depending on the residues surrounding the potential cut site [61]. It was shown there, and by Pan et al. [62], that trypsin cleaves at arginine with greater efficiency than at lysine, owing to the stronger interaction of R with the enzyme’s active site [64]. Under conventional digestion conditions, we see that peptides terminating in arginine are generated with greater frequency than those with a C-terminal lysine. Surprisingly, the enhanced digest at high temperature with added calcium demonstrated equivalent cleavage efficiency at arginine vs. lysine, suggesting a reduction in the bias towards arginine cleavage under the enhanced conditions. We speculate that lysine may be able to interact more strongly with the active site of trypsin due to weakening of the intramolecular H-bonding network within the catalytic pocket [65] at the elevated temperature, while the calcium ions may conserve the conformation sufficiently to maintain high selectivity for lysine and arginine [45]. We also observe from the relative intensity of miscleaved peptides that the enhanced digestion protocol showed a greater degree of cleavage of lysine and arginine when adjacent to acidic or basic residues. These sites are known to electrostatically inhibit the interaction of adjacent tryptic sites with the enzyme’s catalytic pocket, suggesting that the elevated temperature, the calcium-assisted protocol increased the cleavage efficiency of ‘difficult to digest’ cut sites, further supporting enhanced enzyme specificity in the optimized rapid digest.

## 5. Conclusions

The present study characterizes the relationship between trypsin stability and bottom-up MS results at the qualitative and quantitative level. We demonstrated that trypsin de-activation follows second-order kinetics until the rate of denaturation exceeds that of autolysis. Further analysis of these kinetics models based on cumulative activity facilitated the estimation of optimal proteome digestion conditions for a bottom-up workflow. Subsequent bottom-up LC-MS/MS comparisons with relative peptide quantitation against a control showed that the number and abundance of fully cleaved peptides was optimized following a 1 h digest at 47 °C with 10 mM added calcium ions. The corresponding proteome coverage was equivalent to that of a conventional digest. However, the optimized rapid digest demonstrated enhanced tryptic cleavage specificity by way of reduced abundance of semi-tryptic peptides and greater cleavage efficiency at conventionally inhibited cut sites. In summary, we have shown that a 1 h digest at 47 °C with 10 mM added calcium ions provides a more complete and more specific digest than the conventional overnight approach while also increasing throughput. This study employed unmodified, TPCK-treated trypsin. Commercial preparations of modified trypsin may also include calcium in their preparations. While preliminary BAEE assessment of enzyme activity and stability for modified trypsin showed results equivalent to the calcium-enhanced conditions, it is not yet known to what extent the enhanced stability can be attributed to the modified trypsin vs. the addition of calcium. This will be addressed in a future study. We recommend adoption of this efficient, economical, and robust approach to optimize reproducibility, quantitation accuracy and turnaround time.

## Figures and Tables

**Figure 1 biology-11-01444-f001:**
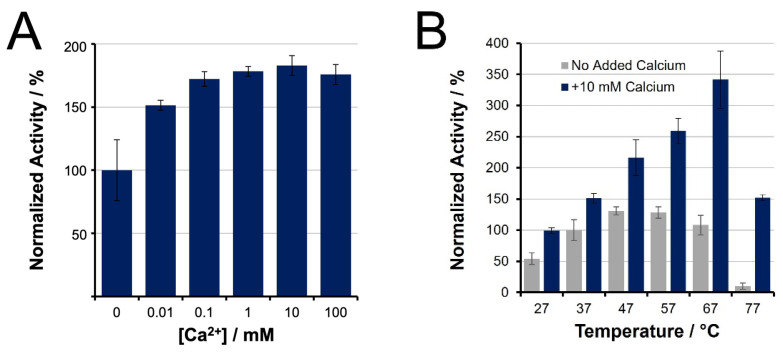
Initial trypsin activity of unmodified, TPCK-treated trypsin, as determined by BAEE assay, and normalized to activity at 37 °C in the absence of calcium. (**A**) Activity increases with addition of CaCl_2_; (**B**) Initial activity at increasing temperatures, showing the benefits of Ca^2+^ for higher temperature incubation.

**Figure 2 biology-11-01444-f002:**
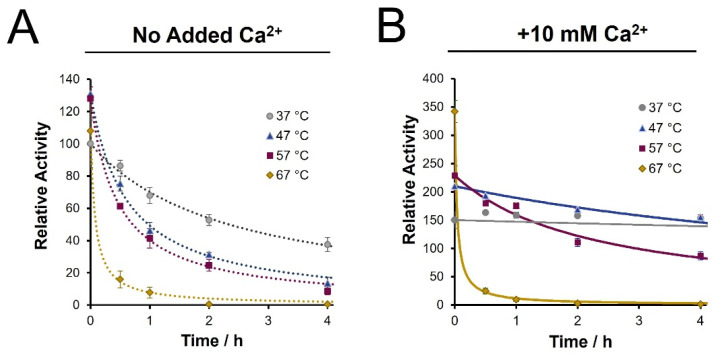
Time course assessment of trypsin activity following pre-incubation (pH 8), as determined by BAEE assay. (**A**) Loss of enzyme activity in the absence of calcium ions is noted at the specified temperatures; (**B**) Inclusion of 10 mM Ca^2+^ increases initial activity and sustains enzyme stability up to a maximum 57 °C. All values are normalized to the initial activity at 37 °C in the absence of calcium. Trendlines represent a fit to a second order kinetics model of the form *A_T_* = *A*_0_/{1 + *ktA*_0_}.

**Figure 3 biology-11-01444-f003:**
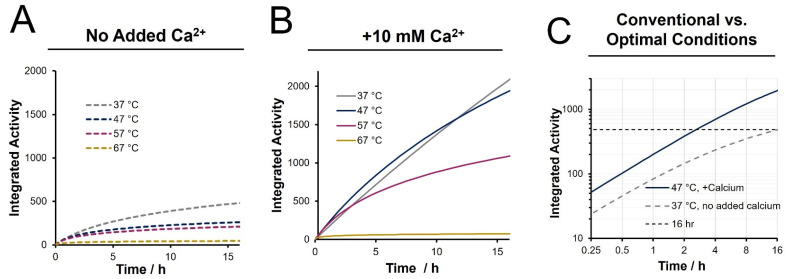
Cumulative activity across a 16 h time course estimated from integrated activity over time (**A**) in the absence of calcium, (**B**) with 10 mM added calcium ions, and (**C**) comparing the proposed optimal condition of 47 °C with 10 mM calcium ions to conventional conditions.

**Figure 4 biology-11-01444-f004:**
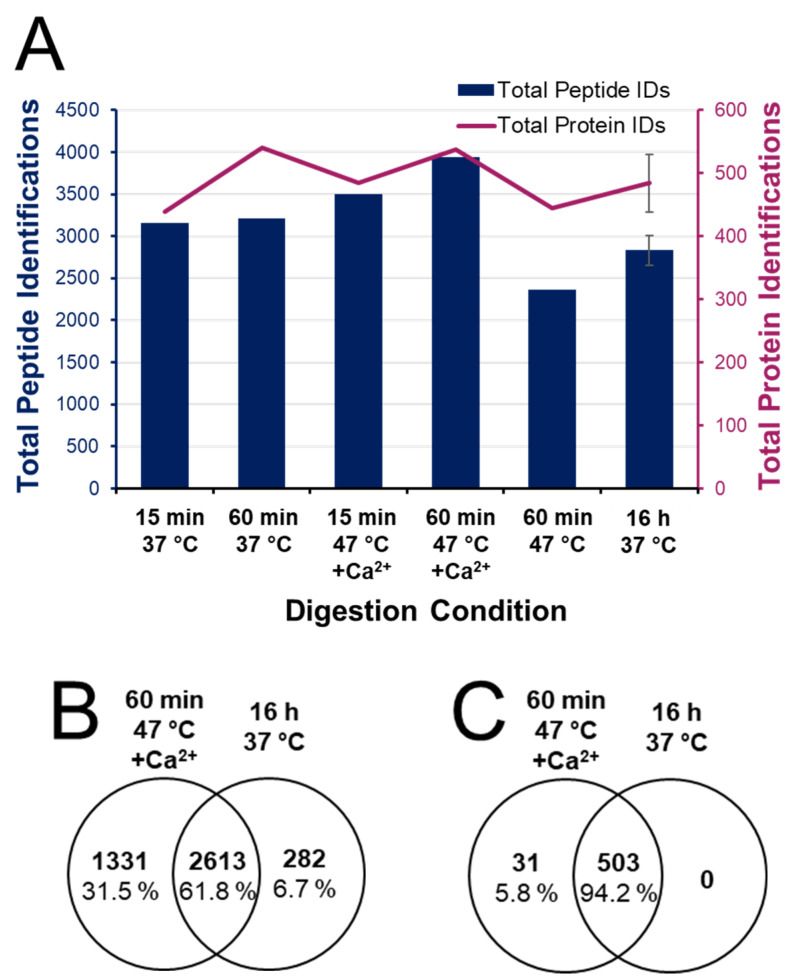
Qualitative bottom-up proteome identifications. (**A**) Total peptide and protein identifications following digestions ranging from 15 min to overnight at 37 or 47 °C, with and without 10 mM added calcium chloride. Venn diagrams of (**B**) peptide and (**C**) protein identifications in the 1 h digest at 47 °C with 10 mM Ca^2+^ are compared to the conventional overnight digest at 37 °C with no added Ca^2+^.

**Figure 5 biology-11-01444-f005:**
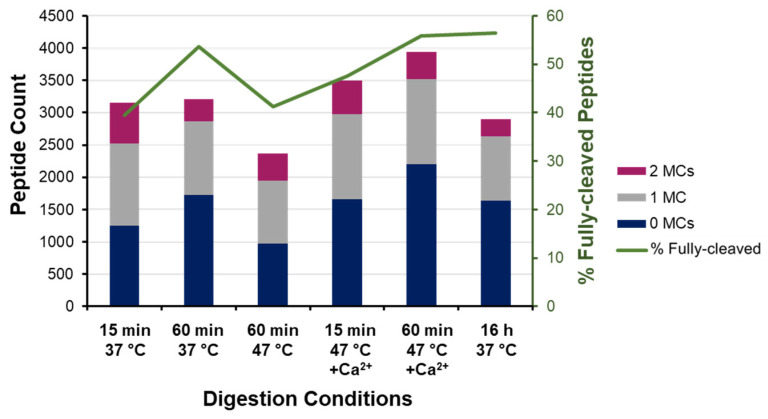
Miscleavage analysis of bottom-up proteome identifications. Absolute count and frequency of fully cleaved and miscleaved peptides identified across digestion conditions.

**Figure 6 biology-11-01444-f006:**
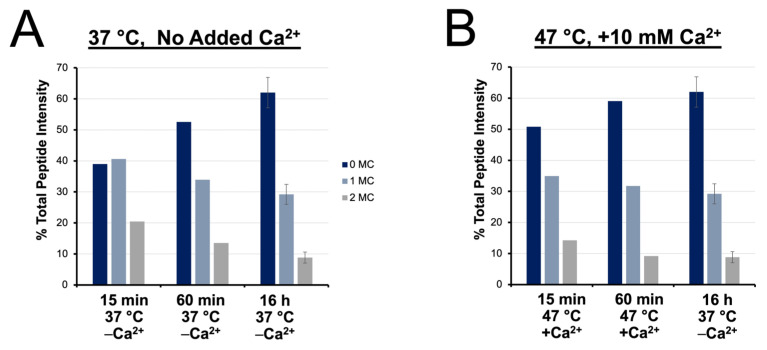
Relative quantitation of fully cleaved, singly miscleaved, and doubly miscleaved peptides. Time course of digestion completion, inferred by % total peptide intensity contributed by fully cleaved, singly and doubly miscleaved peptides at (**A**) 37 °C with no added Ca^2+^ and (**B**) 47 °C with 10 mM added Ca^2+^ compared to a conventional overnight digest.

**Figure 7 biology-11-01444-f007:**
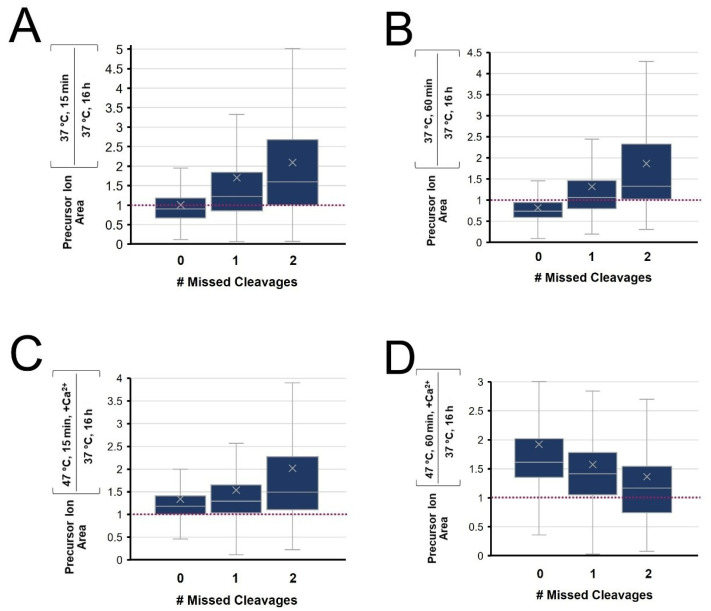
Tukey Box-and-Whisker plots of the distribution of relative peptide peak areas following digestion at (**A**) 15 or (**B**) 60 min, each at 37 °C with no added calcium or at 47 °C with 10 mM added calcium for (**C**) 15 or (**D**) 60 min. The relative peak intensity is compared to a conventional overnight digest at 37 °C.

**Figure 8 biology-11-01444-f008:**
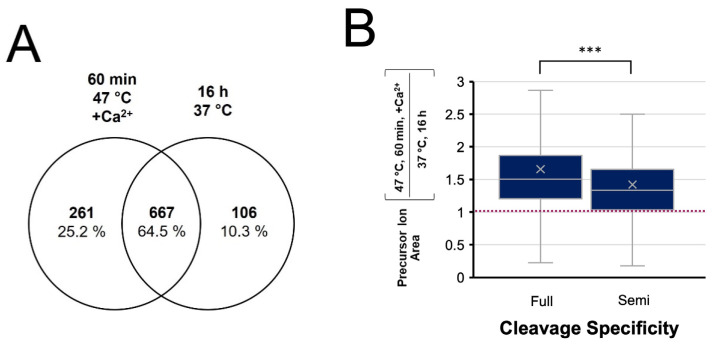
Evaluation of cleavage specificity in the 1 h digest at 47 °C with Ca^2+^ compared to a conventional overnight digest at 37 °C. (**A**) Venn diagram of semi-tryptic peptides identified in each digestion condition. (**B**) Tukey Box-and-Whisker plot comparing peptide quantitation for fully tryptic and semi-tryptic peptides, showing a greater relative abundance of fully tryptic peptides, *** *p* < 0.001.

**Figure 9 biology-11-01444-f009:**
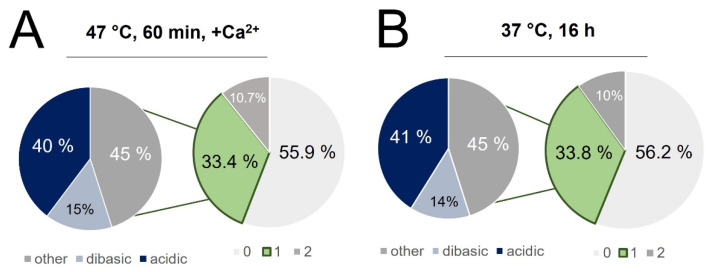
Singly miscleaved peptides identified in (**A**) the rapid digest and (**B**) conventional digest were further classified based on the nature of the missed cleavage site, with 55% of singly miscleaved peptides attributed to acidic or basic residues immediately adjacent to the cut site. (**C**) Motif diagrams highlighting the frequency of acidic and basic residues being within 2 residues of the C-terminal cleavage site. (**D**) Comparison of the slopes correlating peptide intensity in the rapid vs. conventional digest for all peptides, those whose C-terminal cleavage site is adjacent to an acidic residue and those whose C-terminal cleavage site is adjacent to another lysine or arginine (*** *p* < 0.001).

**Table 1 biology-11-01444-t001:** Summary of trypsin de-activation rates at various temperatures in the presence or absence of calcium.

	No Added Ca^2+^			+10 mM Ca^2+^		
Temperature	Initial Activity/%	Rate Constant, k/h^−1^	Calculated Half-Life /h	Initial Activity/%	Rate Constant, k/h^−1^	Calculated Half-Life /h
37 °C	100 ± 15	0.004 ± 1 × 10^−4^	2.4	150 ± 20	0.0001 ± 2 × 10^−5^	51.3
47 °C	130 ± 12	0.013 ± 8 × 10^−4^	0.6	210 ± 17	0.0004 ± 8 × 10^−5^	11.6
57 °C	128 ± 7	0.017 ± 1 × 10^−4^	0.5	229 ± 16	0.0018 ± 2 × 10^−4^	2.4
67 °C	108 ± 20	0.12 ± 8 × 10^−3^	0.03	340 ± 20	0.19 ± 0.01	0.03

## Data Availability

Mass spectrometry proteomics data have been deposited to the ProteomeXchange Consortium (http://proteomecentral.proteomexchange.org accessed on 27 July 2022) via the PRIDE partner repository with the dataset identifier PXD035682.

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
