# Peer review of "Maximizing Cumulative Trypsin Activity with Calcium at Elevated Temperature for Enhanced Bottom-Up Proteome Analysis"

_biology, 2022, doi:10.3390/biology11101444_

Round 1

Reviewer 1 Report

Well written paper with valuable information. 

My only comment is on the sample preparation: you have considerable amounts of CaCl2 in the trypsin solution. Is there any form of sample clean-up before injecting it in the LC-MS or do you switch away the injection front. 

Please comment on that in the manuscript

Author Response

Well written paper with valuable information. 

  1. My only comment is on the sample preparation: you have considerable amounts of CaCl2 in the trypsin solution. Is there any form of sample clean-up before injecting it in the LC-MS or do you switch away the injection front.  Please comment on that in the manuscript

In addition to CaCl2, the digestion buffer contains several additives including urea (1.5 M), DTT (5 mM), IAA (11 mM), TEAB (50 mM), and in these particular experiments, the dimethyl labeling reagents (e.g. formaldehyde, sodium cyanoborohydride). As we noted in the experimental (line 147-148), all samples were subject to reversed phase LC-UV clean-ups prior to LC-MS.

Reviewer 2 Report

In this manuscript, Nickerson and Doucette evaluate ways to improve trypsin digestion for proteomics applications. Given how central the generation of peptides is to bottom-up proteomics, this is an important topic and the findings could have widespread implications. Overall, it is a solid study, well-described and covering multiple different aspects of the central question. The manuscript is also clear and easy to read. A decision of accept with minor revisions would be appropriate for this manuscript.

In the introduction, there is mention that “Counter-intuitively, improved sequence coverage can be obtained through incomplete digestion”. This phenomenon was well explored by Catherine Fenselau’s group in their studies of middle-down proteomics. It would be valuable to include a description and cite some of their work.

The finding that Calcium has such potent effects on trypsin activity could have huge value for the proteomics community and should be implemented into more workflows.

Some specific comments meant to strengthen the manuscript:

Figure 1A.: The data shows that 10 mM has the highest normalized activity, but it does not seem to be statistically significant. Was any testing down with lower concentrations of Ca like 5 mM or 20 mM that would have been better.

Figure 4A. At 16 H 37°C we see a decrease in peptides identified. Why would the peptides identified go down if we are losing tryptic activity?

Figure 8B: This is showing that the full cleavage is statistically higher than the semi cleavage in the 47°C. But this still shows that we had increased semi tryptic peptides increased as well in 47° vs 37°

It would have been interesting to see different calcium concentrations tested. Also, the added control of 37°C with calcium overnight as another sample or control would have been valuable.

In addition, an additional control of known peptides would be valuable. It would have added a bit to the supplemental data to show that we have increased peptide identification and protein coverage with a control mixture than only using the sample.

Author Response

In this manuscript, Nickerson and Doucette evaluate ways to improve trypsin digestion for proteomics applications. Given how central the generation of peptides is to bottom-up proteomics, this is an important topic and the findings could have widespread implications. Overall, it is a solid study, well-described and covering multiple different aspects of the central question. The manuscript is also clear and easy to read. A decision of accept with minor revisions would be appropriate for this manuscript.

  1. In the introduction, there is mention that “Counter-intuitively, improved sequence coverage can be obtained through incomplete digestion”. This phenomenon was well explored by Catherine Fenselau’s group in their studies of middle-down proteomics. It would be valuable to include a description and cite some of their work.

We have cited Fenselau’s work on middle-down proteomics in the revised introduction (line 61-62 + reference 21).

The finding that Calcium has such potent effects on trypsin activity could have huge value for the proteomics community and should be implemented into more workflows.

Some specific comments meant to strengthen the manuscript:

  1. Figure 1A.: The data shows that 10 mM has the highest normalized activity, but it does not seem to be statistically significant. Was any testing down with lower concentrations of Ca like 5 mM or 20 mM that would have been better.

There is no statistical significance between the enhancement in initial activity between 1 and 10 mM calcium. However, we have revisited the question of whether calcium concentration influences trypsin stability over time. As seen in the newly added Supplemental Figure S1 of the revised manuscript, we quantified the rate of trypsin deactivation at 47 oC in the presence of 0, 1, 2, 5, 10, and 100 mM Ca2+. As seen, 5-100 mM Ca2+ provide similar stability. Inclusion of 10 mM Ca2+ maximizes the calculated cumulative activity over longer digestion periods in excess of 1 h. Increasing the calcium concentration (even to 100 mM) has minimal benefit. There is no benefit to reducing calcium below 10 mM.

  1. Figure 4A. At 16 H 37°C we see a decrease in peptides identified. Why would the peptides identified go down if we are losing tryptic activity?

Peptide ID counts (summarized in Figure 4A) can increase or decrease for several different reasons. Even after 16 H, some trypsin activity remains, meaning digestion will proceed. Compared to the shorter digests at 37oC, peptides identified with 1-2 missed cleavages decrease in abundance, while fully cleaved peptides increase (Figure 6A). Thus, a more complete digestion represents a loss in heterogeneity of peptides, and therefore a lower total number of peptides. This is only one explanation; we are currently assessing the degree of peptide modifications from unanticipated ‘side reactions’ that may increase over time.

In the revised manuscript, we have expanded our discussion of Figure 4 (lines 247-255). We examined all miscleaved peptides identified in the conventional digest, to determine if corresponding fully-cleaved peptide segments were present in the rapid digest. Newly added Supplemental Figure S3 shows that miscleaved peptides from the conventional preparation are represented in the rapid digest, but as more completely cleaved segments.

  1. Figure 8B: This is showing that the full cleavage is statistically higher than the semi cleavage in the 47°C. But this still shows that we had increased semi tryptic peptides increased as well in 47° vs 37°

Indeed, the rapid digestion protocol at 47oC demonstrates an increase in abundance of all types of peptides, including both fully cleaved as well as semi-tryptic. What Figure 8B demonstrates is that fully tryptic peptides are even higher in abundance relative to the semi-tryptic peptides.

  1. It would have been interesting to see different calcium concentrations tested. Also, the added control of 37°C with calcium overnight as another sample or control would have been valuable.

(See our response to question 2 above). Considering other additives in the sample, the presence of excess calcium does not present a challenge for sample cleanup and LC-MS analysis. Our goal was to ensure that sufficient calcium was present to maximize trypsin activity over extended digestion, as seen from the cumulative activity plots of Figure 3A and 3B. The ‘control’ digestion (overnight at 37oC) was performed in the absence of calcium as this reflects the normal practice of performing digestion without calcium. Our results enforce the essential role of including calcium to maximize sustained activity. Therefore, including calcium in an overnight, 37oC digestion will result in a more complete digestion compared to the same conditions in the absence of calcium. But it will not yield the throughput benefit of an accelerated digestion protocol.

  1. In addition, an additional control of known peptides would be valuable. It would have added a bit to the supplemental data to show that we have increased peptide identification and protein coverage with a control mixture than only using the sample.

Simple mixtures of standard proteins are often employed to validate alternative digestion approaches based on sequence coverage. However, as highlighted in our introduction, the metric of sequence coverage does not support a complete digest, which is most amenable for high proteome coverage and quantitation accuracy. An increase in peptide identification and protein coverage is evident from multiple examples within the proteome mixture.

Reviewer 3 Report

The manuscript by Nickerson et al described the development of a revised protocol for trypsin digestion in MS application. The authors showed that trypsin digestion at 47 C with calcium for only 1 h can achieve comparable or better cleavage compared to conventional 37 C overnight condition. Elevated temperature and calcium have been studied for trypsin in accelerating digestion or stabilizing it. Based on the results showed in the manuscript, the main advantage of this revised protocol is efficiency. However, the problems such as protein destabilization caused by using higher than physiological temperature cannot be excluded. As a result, it is hard to conclude that this optimized protocol could replace current approach. In summary, despite the considerable amount of information included in this manuscript, this reviewer cannot recommend this paper for publication.

Some other points:

1.     In Figure 1 the authors showed elevated trypsin activity with increased temperature and calcium. A control experiment without trypsin is needed to confirm that the substrate doesn’t degrade at high temperature.

2.     In Figure 6, some data points have error bars, and some don’t.

Author Response

The manuscript by Nickerson et al described the development of a revised protocol for trypsin digestion in MS application. The authors showed that trypsin digestion at 47 C with calcium for only 1 h can achieve comparable or better cleavage compared to conventional 37 C overnight condition. Elevated temperature and calcium have been studied for trypsin in accelerating digestion or stabilizing it. Based on the results showed in the manuscript, the main advantage of this revised protocol is efficiency. However, the problems such as protein destabilization caused by using higher than physiological temperature cannot be excluded. As a result, it is hard to conclude that this optimized protocol could replace current approach. In summary, despite the considerable amount of information included in this manuscript, this reviewer cannot recommend this paper for publication.

The (main) advantages of the revised protocol include:

  • Faster digestion
  • More complete digestion
  • More selective digestion
  • Increased peptides and proteome coverage
  • More economical protocol relative to ‘proteomics grade’ trypsin

Our study also demonstrates that the initial trypsin activity measurements (often used to justify an ‘enhanced’ digestion protocol are a poor indicator of the overall efficiency of a digestion. We therefore provide a mathematical model supporting cumulative trypsin activity as a better indicator of digestion efficiency. Through this model and our experimental results, we demonstrate the importance of including calcium at elevated temperature to maintain tryptic activity over extended periods.

Proteins and proteome mixtures are subjected to elevated temperatures as standard practice in most proteomics workflows. For example, the DTT reduction step performed prior to trypsin digestion occurs at 56oC. Granted, the higher order structure of the protein will be destabilized at elevated temperatures. This is, in fact, the reason why a 67oC digestion is ineffective; not only is trypsin stability lost, but protein precipitation is increasingly evident.

Our results collectively demonstrate that a 47oC digestion produces peptides in greater abundance. Had the proteins been lost to heat-induced precipitation, this would not be the case.

Some other points:

  1. In Figure 1 the authors showed elevated trypsin activity with increased temperature and calcium. A control experiment without trypsin is needed to confirm that the substrate doesn’t degrade at high temperature.

We performed BAEE activity assays in the absence of trypsin (line 112-113 of revised manuscript). The substrate is stable at elevated temperatures.

  1. In Figure 6, some data points have error bars, and some don’t.

 As described in the methods section (lines 144-149), each LC-MS injection included a light labeled experimental condition along with a heavy-labeled conventional digest control. This therefore served as a replicate to infer reproducibility across runs. The error bars on the % Total Peptide Intensity from the conventional digest represent the standard deviation across 5 injections.

Round 2

Reviewer 3 Report

The authors addressed most of my concerns.